# Phospholipid Signaling Is a Component of the Salicylic Acid Response in Plant Cell Suspension Cultures

**DOI:** 10.3390/ijms21155285

**Published:** 2020-07-25

**Authors:** Beatriz A. Rodas-Junco, Geovanny I. Nic-Can, Armando Muñoz-Sánchez, S. M. Teresa Hernández-Sotomayor

**Affiliations:** 1CONACYT–Facultad de Ingeniería Química, Campus de Ciencias Exactas e Ingenierías, Universidad Autónoma de Yucatán (UADY), Periférico Norte Kilómetro 33.5, Tablaje Catastral 13615, Chuburná de Hidalgo Inn, CP 97203 Mérida, Mexico; geovanny.nic@correo.uady.mx; 2Unidad de Bioquímica y Biología Molecular de Plantas, Centro de Investigación Científica de Yucatán (CICY), Calle 43 No. 130, Col. Chuburná de Hidalgo, CP 97200 Mérida, Mexico; arms@cicy.mx

**Keywords:** phospholipid signaling, salicylic acid, hormone signaling, phospholipase C, phospholipase D

## Abstract

Salicylic acid (SA) is an important signaling molecule involved in plant defense. While many proteins play essential roles in SA signaling, increasing evidence shows that responses to SA appear to involve and require lipid signals. The phospholipid-generated signal transduction involves a family of enzymes that catalyze the hydrolysis or phosphorylation of phospholipids in membranes to generate signaling molecules, which are important in the plant cellular response. In this review, we focus first, the role of SA as a mitigator in biotic/abiotic stress. Later, we describe the experimental evidence supporting the phospholipid–SA connection in plant cells, emphasizing the roles of the secondary lipid messengers (phosphatidylinositol 4,5-bisphosphate (PIP_2_) and phosphatidic acid (PA)) and related enzymes (phospholipase D (PLD) and phospholipase C (PLC)). By placing these recent finding in context of phospholipids and SA in plant cells, we highlight the role of phospholipids as modulators in the early steps of SA triggered transduction in plant cells.

## 1. Introduction

Plants and other living organisms face new challenges every day. As such, they have developed efficient strategies to adapt to different types of stress to survive and propagate. The stress response occurs through the activation of signal transduction cascades, which control the physiological and biochemical responses needed for plant. In recent years, phospholipids have been reported not only as components of the plasma membrane, but also as important regulatory lipids involved in the response to stress in plant cells [1,2,3]. In recent decades, the number of advances related to phospholipid signaling has increased. Phospholipid signals are produced and metabolized by several enzymes, such as phospholipases, lipid kinases and phosphatases [4,5,6]. Different approaches, such as genetic manipulation studies, omics studies, lipid analyses, molecular interaction analyses and physiological analyses, have been integrated to determine the function of this lipid signaling pathway [7,8,9,10]. Their results have shown that this pathway participates in the process of plant growth, development and responses to the changes generated by biotic and abiotic stresses [6,11]. Therefore, lipid signaling plays an important role in mediating plant hormone effects [7]. Several key proteins of the phosphoinositide metabolic pathway, such as phospholipases and lipid mediators (phosphatidylinositol 4,5-bisphosphate (PIP_2_), inositol 1,4,5-trisphosphate (IP_3_), phosphatidic acid, (PA) and phosphatidylinositol-4-phosphate, (PI4P)), are involved in processes mediated by plant hormones such as salicylic acid (SA).

SA is a phenolic phytohormone that plays a central role in various physiological processes, in defense against biotic and biotic stress such as the regulation of plant growth and development [12,13,14,15,16]. SA is also involved in the regulation of signaling pathways associated with secondary metabolite biosynthesis pathway [17,18,19]. However, little is known about the possible connection between phospholipid signaling and SA-dependent defense responses. In this review we focus on reviewing the current knowledge of the role of phospholipid signaling in the response triggered by SA. Later, the experimental evidence supporting the phospholipid–SA connection in plant cells is discussed, emphasizing how lipid secondary messengers (PIP_2_ and PA) and related enzymes (phospholipase D (PLD) and phospholipase C (PLC)) involved in the control of SA signal transduction. Finally, it is discussed how the mediation of the phospholipid pathway in SA signaling has an important role in the synthesis of secondary metabolites. Discussions of the biochemistry of these phospholipids, their metabolizing enzymes, and their roles in other plant stress responses can be found in several excellent reviews [7,12,20,21,22].

## 2. Salicylic Acid: An Essential Regulator to Mitigate Biotic and Abiotic Stress in Plants

Salicylic acid (SA; 2-hydroxybenzoic acid) is an endogenous molecule that modulates the response to various types of stress in plants, both biotic and abiotic [12,15,16]. SA belongs to a diverse group of phenolic compounds, generally defined compounds that have an aromatic ring with a hydroxyl group or its functional derivative [23]. The most well-established role of SA is as a signaling molecule in the plant immune response that playing direct or indirect roles in the regulation of many aspects of plant growth and development, as well as cell growth, respiration, stomatal aperture, senescence, seed germination, seedling development thermogenesis, flowering and disease resistance [24]. However, notably, the regulatory role of SA in these physiological functions differs because the basal levels of SA vary between different plant species, the stage of development in which they are found, the type of tissue analyzed and exposure to (a) biotic stress. In the following section, several lines of evidence are reviewed that maintain that in SA signaling there are effector proteins that mediate the function of SA under biotic stress, while under abiotic stress SA causes changes in the metabolism of plant cells.

### 2.1. Salicylic Acid Signaling through SA-Binding Protein

To elucidate the mechanisms through which SA induces these responses, several putative effector proteins have been identified. In this sense, multiple investigations via biochemical and genetic approaches have revealed that SA can act through multiple effector proteins in plants called SA-binding proteins (SABPs) [25,26,27,28]. Several SABPs have currently been purified from extracts in different organs of plants or suspension cells. For example, inhibition of SABP1, which was purified from tobacco leaves and suspension cells as a cytosolic (peroxisomal) catalase (SABP1-CAT1) mediated by SA may result in the H_2_O_2_ burst seen during the hypersensitive response (HR) leading to the activation of defense proteins [25,29]. Additionally, in *Arabidopsis*, an SAPB1 protein named CAT2 has been identified in leaves pretreated with SA [30]. However, how SA binds SABP1 remains unclear.

Another protein identified as SA-binding protein 2 (SABP2) from tobacco exhibits high affinity for SA (Kd = 90 nM). SABP2 has methyl salicylate esterase (MeSA) activity, converting MeSA into SA and playing a crucial role in the activation of systemic acquired resistance in response to plant pathogens. Interestingly, T-DNA insertion lines defective in expression of a pathogen-responsive SA methyltransferase gene are completely devoid of induced MeSA production but increase systemic SA levels and develop SAR upon local *P. syringae* inoculation. Therefore, MeSA is dispensable for SAR in *Arabidopsis*, and SA accumulation in distant leaves appears to occur by de novo synthesis via isochorismate synthase [31]. In addition to SABP2 proteins in tobacco and *Arabidopsis*, SABP2 ortholog have been characterized in poplar and potato showed an inhibition of esterase activity in response to SA in vitro [32,33]. In addition, SABP3, a chloroplastic carbonic anhydrase was identified in tobacco as β-carbonic anhydrase [20,25]. This protein also plays a role in the HR and, interestingly may have antioxidative properties [25]. Recently, other SABPs have been identified and validated in *Arabidopsis*, such as thimet oligopeptidases (TOPs), glyceraldehyde 3-phosphate dehydrogenase (GADPH), glutathione S-transferase, α-ketoglutarate dehydrogenase and thioredoxins [25].

Research on the role of SABPs in response to abiotic stress, is lacking. Recently, Li et al. (2019) demonstrated that overexpression of *LcSABP* (an orthologous gene of SABP2) from *Lycium chinense* is involved in the regulation of the drought stress response through a SA-dependent defense pathway in transgenic tobacco plants [34]. However, although several approaches have uncovered multiple SABPs, none of these proteins seem to function as typical receptors that mediates activation of the stress response in plants. Therefore, other research groups have proposed different models of the perception of SA through the master regulator NON-EXPRESSOR OF PATHOGENESIS-RELATED GENES 1 (NPR1) or the negative regulators NPR3/NPR4 in SA-induced immune responses [35,36,37,38]. For example, Ding et al. (2018), reported that, rather than acting as a regulator of NPR1 protein stability, NPR4/NPR3 function as cotranscriptional repressors that regulate SA signaling independently of NPR1, demonstrating that the recognition of SA can occur through the contrasting roles of SA-binding NPR proteins [35]. All this evidence reported by different research groups and other research is various aspects are complementary, ultimately allowing us to establish a molecular scenario in which SA signaling occurs under biotic stress: (1) SA alone cannot induce a response in plants, but must orchestrate the response through protein components that transmit the signal during the plant defense response; (2) The presence of SABPs indicates that there exists a mechanism independent of NPR1 that plants must control the transition of the stress response towards their cellular metabolism; and (3) The recognition of SA by NPR1 receptors or their paralogs NPR3 and NPR4 would be under strict regulation, with contrasting roles that depend on a wide range of SA concentrations, which would partially explain the different responses observed between different plant species.

### 2.2. SA as a Mitigator of Abiotic Stress

Abiotic stress is one of the most challenging threats to agricultural systems and productivity of crop plants. It is difficult to estimate the effects of abiotic stress on crop production, but there is a substantial impact on plants. Phytohormones constitute a solid tool to improve the effects of stress on crops of high commercial demand. In particular, SA has been shown to improve plant tolerance to abiotic stresses such as metals, ozone, UV radiation, chilling, salinity, heat, cold and drought [see review 40]. Compared with those on biotic stress, studies on SA-modulated abiotic tolerance have focused on the physiological level, demonstrating that the protective effect of SA is associated with the antioxidant system, the accumulation of osmolytes, secondary metabolites or even an increase in mineral nutrients [39]. It is clear that SA is part of a complex signal transduction network, and its protective effects may differ by the species and developmental stage of the plants, their genetic background (dicotyledonous vs. monocotyledonous plants), the concentration of exogenous SA and its endogenous level in a given plant. In this context, the use of plants cells as a simplified and amenable experimental model has allowed the study of SA-induced signaling mechanisms that would be too complex in plant tissues or organs [40]. Several lines of evidence using plant cells suggest that signaling pathways components such as reactive oxygen species (ROS), abscisic acid (ABA), Ca^2+^ and phospholipids interact with the SA signaling pathway [2,23,24,41]. However, compared to those involved in biotic stress, the molecular mechanisms involved in abiotic stress in response to SA remain poorly explored. Therefore, we focus on discussing the regulation of some major players in the signaling pathway mediated by phospholipids in response to SA with a focus on plant cells.

## 3. Phospholipid Signaling in Plants

Phospholipids are the main and vital components in all membranes in eukaryotes.-Most membrane bilayers comprise structural membrane lipids, such as phosphatidylethanolamine (PE) phosphatidyl choline (PC) and phosphatidylserine (PS), which together account for 70%–80%, followed by phosphaytidylglycerol (PG) and phosphatidylinositol (PI) (5–10%). Phospholipids are composed of two fatty acid tails, glycerol, a phosphate group and a polar head group (Figure 1). Phosphoinositides (PPIns) is a term used to describe the seven types of phosphorylated PIs and are the best examples of lipids with important regulatory functions in various cellular processes, including the control of membrane trafficking, cytoskeletal remodeling, ion transport and signal transduction [9,42]. In this context, lipid signaling in the membrane is the perfect mechanism for information transmission throughout the plasma membrane, cytosol and other organelles, particularly the nucleus [43,44].

Phospholipid-mediated signaling involves the generation of messengers by phospholipases and/or lipid kinases [7]. Phospholipases are classified according to the site of phospholipid cleavage and the nature of their products: A1, A2, C and D (PLA1, PLA2, PLC and PLD, respectively) [40,45] (Figure 1). Some phospholipases are characterized by their strict substrate preference. For example, PLCs are either phosphoinositide-specific PLCs (PI–PLCs) or nonspecific PLCs (NPCs). PI–PLCs hydrolyze PIP_2_, generating two second messengers—IP_3_ and diacylglycerol (DAG). In mammals, IP_3_ binds to specific calcium channels, triggering the release of calcium from internal stores (such as the endoplasmic reticulum) into the cytosol. No IP_3_-binding channels have been identified in terrestrial plants [46]; however, reports have indicated that IP_3_ can be phosphorylated into higher phosphorylated forms, such as inositol–hexakisphosphate (IP_6_; also known as phytic acid) by inositol kinases (IPKs) [7,42]. Instead, DAG in plants is rapidly phosphorylated by DAG kinase (DAGK) to produce PA, which has been shown to exert several regulatory functions [47]. PA can also be formed by the hydrolysis of PC, PG or PE, via the action of PLD [48] (Figure 2). The specific phospholipids at various intracellular locations is essential for the regulation of a range of important cellular processes. It has been well characterized that the recruitment of proteins to the plasma membrane is fundamental to initiate and regulate signal transduction events [49].

### 3.1. Phospholipid Signaling-SA Connection

As reviewed in previous section, while protein components are essential for SA signaling, important roles for lipids and their enzymes have also been revealed to be an essential part of SA signal transduction. The importance of some components of phospholipid signaling in response to SA can be briefly illustrated by a few examples. In *Arabidopsis* suspension cells, Krinke et al. (2009) demonstrated that SA stimulation led to rapid activation of PLD; however, when PLD activity is modified due to the presence of primary alcohols and consequently reduces PA levels, transcriptomic changes stimulated by SA are affected [50]. Interestingly, in the same model, it was shown that PI–PLC substrates and products participate in SA-triggered transcriptomic remodeling [40] Therefore, these two studies clearly show a connection between lipid signaling and SA in controlling gene expression. In contrast, other research groups have contributed by focusing on enzyme activity in vitro, in vivo lipid labeling in different plant cell cultures or the generation of mutant plants to establish the role of phospholipids in SA signaling.

### 3.2. SA Differentially Alters PLC and PLD Activity in Plant Cells

Plant cell cultures are useful tools for investigating of physiological phenomena such as cell proliferation and differentiation in plants. In such experimental systems the environment should be completely controllable, and the population should be homogeneous. Furthermore, if the model is well characterized it can provide a suitable system for the study of intracellular signaling in plants.

However, to the best of our knowledge, the consequences of age-related changes in phosphoinositide metabolism regarding plasma membrane signaling have not been addressed in detail. For this reason, our research group carried out a study focused on two key phospholipases with relevant functions in plant signaling, PI–PLC and PLD, to evaluate the effect of SA on the enzymatic activities during the culture cycle of *C. chinense* cell suspensions. In this context, our first hypothesis was that, during the cell growth cycle, constant metabolic changes occur, and phospholipase function is probably an effective way to control these processes. To this end, we investigated whether different concentrations of SA (25–200 µM) for 30 min affected the activity of PI–PLC and PLD in cell cultures. In vitro assays with a radiolabeled substrate ^3^[H]-PIP_2_ and ^3^[H]-PC, were using in both assays to measure the formation of IP_3_ for PLC activity and choline for PLD activity. Our data showed that SA treatment differentially modified PI–PLC and PLD activities in a dose-dependent manner. Interestingly, PI–PLC and PLD activity was stimulated mainly by 25-μM SA [51]. In contrast, with 200 µM SA, PLC activity was inhibited, but not PLD activity [52]. Activation of PLD in response to SA was also observed in cell suspensions of *Arabidopsis* treated with 250-μM SA at 45 min after SA treatment [46]. Taken together, these results show that SA uses phospholipid-mediated signaling machinery for signal transduction, suggesting finely tuned regulation of this hormone mechanism. Furthermore, the identification of multiple isoforms of PLC and PLD, as well as their weak expression patterns of enzymes according to DNA chip technology in *Arabidopsis*, *Oryza sativa* and *Glycine max*, suggest different regulatory mechanisms in response to exogenous phytohormone treatment [25,53].

## 4. Association between Phosphoinositide Lipid Second Messengers and SA

### 4.1. Physiological Role of PIP_2_ in Plants

PIP_2_ is synthesized by phosphorylation at the D-5 position of the inositol ring of PI4P by phosphatidylinositol 4-phosphate-5-kinase (PI4P-5 K). In plants, PIP_2_ represents less than 1% of membrane phospholipids and performs various key cellular functions far beyond its role as a precursor to IP_3_ and DAG. Among the widely reported functions include its role in actin cytoskeleton organization, dynamic recruitment of cytoskeletal proteins, signaling to the plasma membrane, intracellular vesicular trafficking secretion and stimulation of PLD (for review see [47,54,55]. For instance, the levels of PIP_2_ change in response to environmental stresses, including wounding [56], salt or osmotic stress [57,58,59] and heat [60]. In recent years, PIP_2_, PI4P and its synthesizing enzyme, PI4PK, have been intensively studied in plant cells due to their roles in signal transduction, not only as precursors of second messengers, but also as membrane-bound regulators of signaling proteins [61]. In this sense, several proteins regulated by PI binding have been identified; in some studies, these proteins have been termed PI “modulins” [21,62] *A. thaliana* root hairs and pollen tubes [63] and as well as hypocotyls of *Brassica oleracea* [64] have been used as models to study PIP_2_. Various reports have indicated the presence of PIP_2_ in the membrane microdomains of pollen tubes or in the plasma membrane of tips of root hair cells [65,66,67]. Disruption of PIP_2_ (biosynthesis or hydrolysis) interferes with vesicle trafficking and affects pollen tube growth, supporting a role for PIP_2_ in the regulation of pollen tube growth [67,68,69]. The involvement of PIP_2_ in mediating plant adaptations to stress, such as salt or osmotic stress was evaluated in *Arabidopsis* plants by Dewald and coworkers (2001) [57]. These authors reported the accumulation of PIP_2_ in plants, suggesting that PIP_2_ plays an important role in the stress response by modulating the activity of cytoskeletal-associated proteins and/or modifying vesicle trafficking in response to osmotic stress.

### 4.2. Role of PIP_2_ in SA Signaling

The importance of hormones in PIP_2_ turnover can be briefly illustrated by a several examples.

Tejos et al. 2014 reported that exogenously applied auxin modulates PIP_2_ levels and that the PI4P 5-kinases PIP5K1 and PIP5K2 are redundantly required for polar localization of specifically apical and basal cargoes, such as PIN-FORMED transporters (which are essential for directional movement of the auxin) for the plant hormone auxin in the plasma membrane polar domains in *Arabidopsis* root cells [70]. In *Arabidopsis* suspension cells radiolabeled with ^33^P, [50], showed that SA activated a type-III phosphatidylinositol-4-kinase (PI4 K) accompanied by a rapid increase in the labeling of PI4P and PIP_2_. In this context, it is important to understand how SA could activate PI4 K in other plant models species. In this sense, Sasek et al. 2014 [4] showed that a double knockout mutation of two isoforms of PI4 Kβ1β2 triggers SA signaling. In 2011, our research group performed an in vitro enzyme assay of microsomes extracted from *C. chinense* Jacq. cells 30 min after SA treatment. The results showed that SA induced an increase in lipid kinase activities leading to PI4P and PIP_2_ and a decrease in PI–PLC activity [52]. Similarly, a SA-induced increase in PIP_2_ content has been reported in *Arabidopsis* cells [40]. The increased PIP_2_ level in these studies could be seen as a way to supply substrates for the action of PI–PLC; however, since the activity of PLC is inhibited by SA, it is inferred that the accumulation of PIP_2_ can stimulate PIP_2_-dependent PLD isoforms or other PIP_2_-PLC isoforms that may have been differentially affected by SA [50,52]. These findings strongly suggest that PIP_2_ is a component of the SA signaling pathway and that the activation of this molecule is necessary for the SA response. Furthermore, it cannot be excluded that protein kinases act upstream in response to SA. For further insight, the identification of PIP_2_-binding proteins crucial to the SA-signaling cascade is imperative.

### 4.3. PA Is Involved in SA Signaling

Phosphatidic acid (PA) is a central intermediary in glycerolipid biosynthesis and a potent lipid mediator involved in the regulation of various cellular processes such as lipid metabolism, signal transduction, cytoskeletal rearrangements and vesicular trafficking [1,48,71,72]. The concentration of this phospholipid in *Arabidopsis* leaves ranges from 50 to 150 µM, representing 1% of total phospholipids [11,71]. PA is described as an important signaling molecule in plants and animals. The formation of this lipid can increase as a rapid response (minutes) and is transiently generated in response to biotic and abiotic stresses. This signal-induced PA is produced via two distinct enzymatic pathways. The first route is accomplished in a two-step enzymatic process that involves the generation of diacylglycerol (DAG) from inositol phospholipids catalyzed by PLC, followed by the production of PA through the phosphorylation of DAG by DGK [73,74]. In the other route, PA can be formed directly through the hydrolysis of structural phospholipids such as phosphatidylcholine and phosphatidylethanolamine by PLD, mostly contributing to the formation of this molecule [1,73,74,75]. The signal is attenuated by the action of PA phosphatase or by the conversion of PA into DAG pyrophosphate (DGPP) by PA kinase [76]. The participation of lipid metabolism and SA signaling has been evident in recent years. The identification of SABP2 (a protein with lipase-like activity) and its involvement as an essential component in SA triggered signaling is a clear example of this. Recently, the biologic importance that PA could have in hormone signaling has gained increase amounts of attention. Interestingly, several independent studies have determined that the PA response is biphasic during the host–pathogen interaction in cells/tissues of tobacco and *Arabidopsis* as well as in response to treatment with biotic and abiotic elicitors such as SA [72,77].

To understand the role of PA in signal transduction and hormone responses in depth, several research groups have used knockout mutants of the PLD genes or application of 1-butanol (an antagonist of PLD-dependent PA production), which results in an inactive phosphatidyl alcohol [78,79,80]. PLD-derived PA is involved in regulating various phytohormone signaling pathways, including those of abscisic acid (ABA), gibberellic acid, ethylene and cytokinin [71,81,82]. However, few studies have investigated the molecular mechanism that connects PA with the SA pathway. In this regard, studies in plants have shown that SA increases PA levels during systemic responses [83] and stomatal closure [45] and that exogenous PA can prevent the disruption of actin filaments caused by SA [84]. However, the mechanism of PLD-derived PA involvement in SA signaling remains unclear. In this regard, Janda et al. 2015 suggested that n-butanol, a primary alcohol that modulates the activity of PLD, is involved in the transport process of NPR1 to the nucleus in *A. thaliana* transgenic plants [78]. Another example is that SA-induced PLDœ-PA signaling mediates NADPH oxidase RbohD activation and ROS production, suggesting that PLDœ activation is an important component that functions downstream of SA [75]. Additionally, using microarray analysis, Krinke et al. 2009 investigated whether genes that responded to SA via PLD-produced PA were inhibited in the presence of 1-butanol [50]. Their results revealed 97 genes that responded to SA, among which some encoded transcription factors and were PR genes (e.g., NPR1, NIMIN1, NIMIN2, WRKY38, WRKY66) [50,75,78]. Additionally, our group observed an increase in PA during SA treatment in the cells of *C. chinense*; in contrast, 1-butanol caused those levels stimulated by SA to decrease [85]. Taken together, these results suggest that PLD-derived PA is an important mechanism in SA signaling.

### 4.4. PA-Binding Proteins Involved in Hormone Signaling

As discussed above, the activation of PLC or PLD can lead to PA production. One key PA action as a signaling molecule occurs via its direct interaction with proteins. However, how does PA interact with its effector proteins and how is its specificity likely achieved? Work by Kooijman et al. (2009) suggested that PA undergoes double deprotonation and acquires a double negative charge; thus, the formation of the hydrogen bond between the PA phosphate group and primary amino group occurs later for a positively charged amino acid residue on the protein, such as lysine or arginine [86]. Furthermore, an increase in the electrostatic interaction of PA is necessary for its specific binding to other phospholipids present within the plasma membrane [74,87,88]. The PA-protein interaction can have two effects: modulation of catalytic activities and intracellular distribution. Substantial progress has recently been made in understanding how PA interacts with various proteins and how it modulate activities in signaling events. In this sense, excellent reviews have reported that different proteins may be possible targets for PA. Among them are protein kinases and phosphatases, lipid kinases, ion channels and NADPH oxidase [48,71,74,76,89,90]. Regarding hormone signaling, direct molecular targets of PA have been identified in abscisic acid (ABA), gibberellic acid, ethylene, brassinosteroid (BR) and cytokinin signaling pathways [71]. In the ABA responses, PLDœ1-derived PA has been shown to bind to ABI1 PP2C, a protein phosphatase that negatively regulates the intracellular response of ABA [91]. The amino acid residue required for the PA-ABI1 interaction has been identified and resides in the N-terminal region of ABI1 [71,91]. PLDœ1-derived PA has also been shown to interact with and stimulate NADPH oxidase. For example, Zhang et al. (2009) showed that PLD-PA is involved in NADPH-oxidase (isoform RbohD) regulation, especially by PA-RbohD physical interactions. The PA binding site in RbohD is located in the cytosolic region between the two EF hands and N-terminus [92]. The role of those proteins has been well described in studies on the response to ABA; however, studies focused on the identification of some PA binding proteins in response to SA are scarce. There is a study published by Matousková et al. 2014 demonstrating that the negative effect of SA on actin dynamics in *Arabidopsis* seedlings was abolished by binding of the capping protein (CAP) with PA [84]. Research into the exact functions of PA in SA-triggered signaling is one of the most interesting frontiers of research in plant cells.

## 5. Salicylic Acid, Phospholipid Signaling and Secondary Metabolites in Plant Cells

Most research on SA has focused on its role in plants in response to biotic and abiotic stresses. However, the SA signal spreads to other parts of the plant to induce multiple defense responses, including the production of certain classes of secondary metabolites. Thus, it is important to study signal transduction related to stress conditions because it would help in the development of strategies for the commercial production of target compounds that activate or suppress certain metabolic pathways. Much effort has been spent to understand the cascades of reactions that result in the formation and accumulation of secondary metabolites in plant cells in response to SA. In this sense, it has been shown that signaling molecules lead to genetic expression and biochemical changes in a specific metabolic pathway [93,94]. Additionally, a whole-genome approaches such as metabolic and transcriptomic profiling in the roots of *Arabidopsis* [95], in the seeds of *Triticum estivum* L. [96], in *Rehmannia glutinosa* hairy roots [94] and in *Scutellaria baicalensis* [97], have been used to determine the genes specifically regulated in response to SA treatment. Extensive studies have indicated that SA is an effective elicitor of secondary metabolites in various plant species. In SA-treated *Linum album* cell cultures, podopyllotoxin production was 3-fold higher than that in control cultures [98]. Ginsenoside in *Panax ginseng* adventitious roots [99], xanthonescadensin G and paxanthone in *Hypericum* spp. suspension cells and hairy root cultures [100], and stilbene in suspension cell cultures of *Vitis vinifera* [101] can be induced by SA.

These results suggest that the accumulation of secondary plant metabolites by SA could modulate the signaling network associated with the secondary metabolism biosynthesis pathway. For example, Hao et al. 2014 reported that intracellular H_2_O_2_ elicited by SA is a secondary messenger of signal transduction that promotes phenylalanine ammonia–lyase (PAL; the first enzyme involved in the accumulation of phenolic compounds) activity and participates in rosmarinic acid accumulation in *Salvia miltiorrhiza* cell cultures [102].

Currently, the SA pathway may be associated with the phospholipid signaling system. Few reports, however, have linked the components of phospholipid signaling activation with elicitation by SA with respect to secondary metabolism. For example, Vasconsuelo et al. (2003) suggested that IP_3_ signaling is involved in chitosan-induced accumulation of anthraquinone synthesis in *Rubia tinctorum*, but the stimulation decreases with the PLC antagonist neomycin and U73122 [103]. In 2014, Ruelland et al. reported that PI–PLC products (IP_3_ or DAG) and substrates (phosphoinositides) participate in SA-triggered transcriptomic remodeling in *A. thaliana* suspension cells [40].

To further understand the role of phospholipid signaling and secondary metabolism in response to SA, our group used in vitro cultures of suspension cells of *C. chinense* Jacq. This crop species has high commercial potential and generates high-economic-value metabolites including capsaicinoids and vanillin.

The exposure of *C. chinense* suspension cultures to SA leads to an increase in the accumulation of vanillin, the phenolic moiety of capsaicin [104]. This response is preceded by an increase in the activity of PAL, a key enzyme involved in the phenylpropanoid pathway in *Capsicum* and requires the participation of phospholipid signaling [19,52]. However, it is unclear whether the role of phosphoinositide-dependent pathways controls PAL gene expression in *C. chinense* cells. Recently, in 2019, we reported the role of the PI–PLC pathway in the transcriptional regulation of *CchPAL1* and *CchPAL5*, which are putative PAL genes in *Capsicum* (Figure 3). We observed that an increase in phosphoinositide levels appears to be important in the *C. chinense* SA-specific response, with PI–PLC signaling contributing mainly to a common SA response [41]. In contrast, when PLD/PA synthesis is blocked by 1-butanol in cells stimulated with SA, the transcriptional regulation of *CchPAL1* and *CchPAL5* in response to SA does not involve the PLD pathway.

## 6. Conclusions and Perspectives

In this review, we summarized various reports describing the roles of plant phospholipids as signaling molecules involved in the SA response. Increasing amounts of evidence from biochemical and genetic studies suggests that phospholipid signaling plays an important role in biotic and abiotic stress. Interestingly, as demonstrated or inferred by multiple studies, components of phospholipid signaling such as PA, PIP_2_ and PI orchestrate an amplification of the SA signal leading to downstream defense responses involving protein–protein and lipid–protein interactions. However, due to the intricate complexity of phospholipid signaling and the experimental conditions (in vitro versus in vivo) in various systems, there are still questions about how signal transduction by SA could occur in plant cells. Future studies using combinations of genetic and cellular approaches could help identify direct lipid–protein interactions in a (sub) cellular context. Likewise, tools such as metabolomics could provide a global perspective of the changes in secondary metabolism in plants, which will allow us to show the impact that each of the components of phospholipid signaling makes on the SA molecular scenario. In particular, the following questions are important to clarify in the lipids-SA connection in plants: (i) Do the components of the NPR1 independent pathway interact with phospholipid mediators? If this happens, what would that molecular mechanism be like in cell, tissues or organs in plants? (ii) In the cellular context, is the spatial location of the components of phospholipid-mediated signaling (phospholipases, substrates/products) key to modulating the response to SA? (iii) An intriguing aspect of lipid signaling regulation may be post-translational modifications and their role in modulating the activities of PLC and PLD in response to SA.

In conclusion, phospholipid signaling is a component of the response to SA in plants and there are still potential opportunities for future research focused particularly on the role of SA with phospholipid signaling. The data generated in future research will allow to develop of strategies based on the exogenous application of SA to improve plants stress responses, increasing the quality and production of crops.

## Figures and Tables

**Figure 1 ijms-21-05285-f001:**
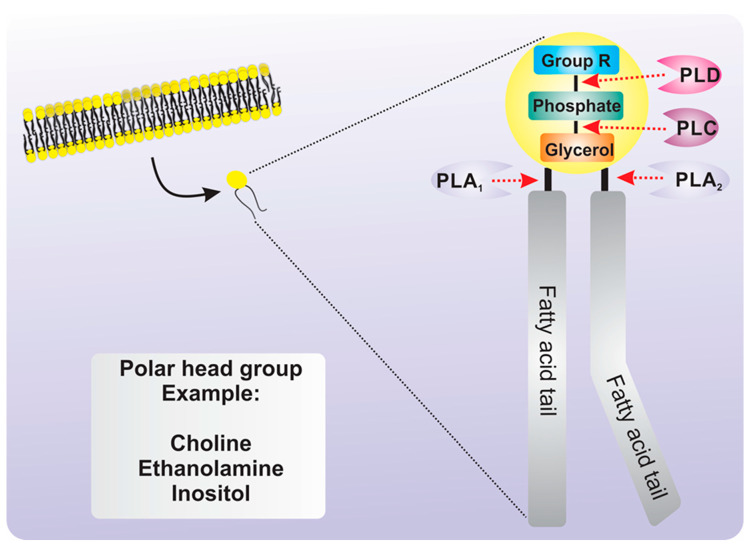
Phospholipid hydrolysis sites by phospholipase A1 (PLA1), phospholipase A2 (PLA2) phospholipase C (PLC) and phospholipase D (PLD). Figure shows a representation of a phospholipid with the three major phospholipases involved in its hydrolysis and the bond hydrolyzed by them in glycerophospholipids with an arbitrary composition of fatty acids. The hydrolysis site is indicated in dashed red arrows. Solid black arrows indicate a phospholipid of plasmatic membrane.

**Figure 2 ijms-21-05285-f002:**
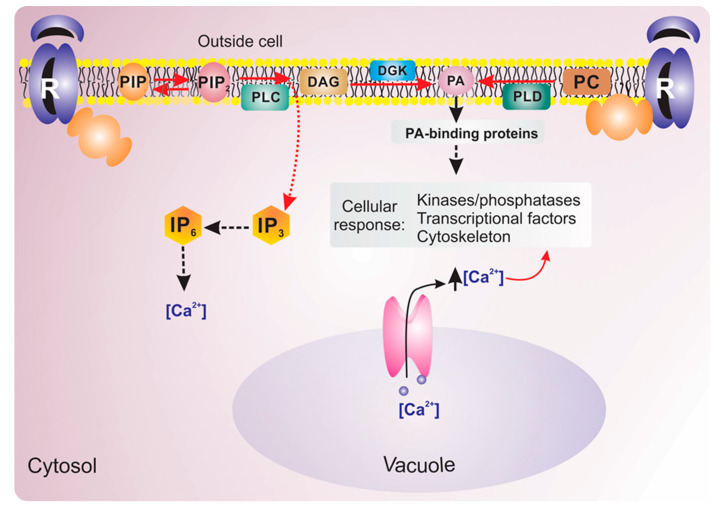
Diagram of phospholipid signaling in plants. PLC can hydrolyze PIP_2_, which generates membrane-bound diacylglycerol (DAG) and IP_3_. While DAG is rapidly phosphorylated by DGK to form the signaling lipid PA which can also be generated by PC hydrolysis by PLD (solid red arrow). IP_3_ diffuses into cytosol (dashed red arrow) where is converted to IP_6_ for which several new signaling functions are emerging. Dashed black arrow indicate regulation (either directly or indirectly) of downstream targets. Solid black arrow indicates calcium release from vacuole.

**Figure 3 ijms-21-05285-f003:**
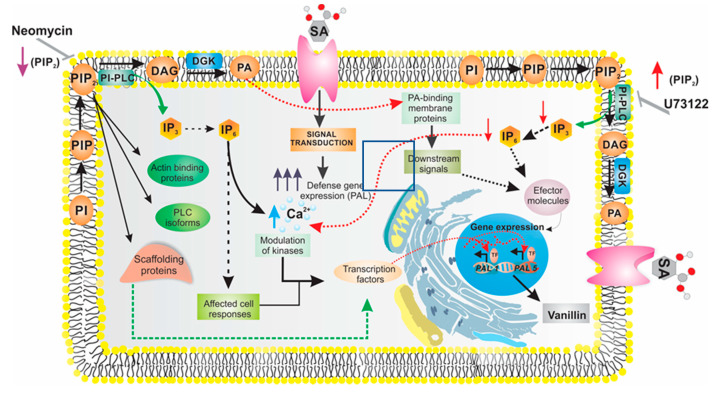
Working model of phospholipid signaling and SA role in the biosynthesis of the secondary metabolites in *C. chinense* cell suspensions. In this model, we propose that SA near the plasma membrane may be identified by a receptor and activate a signaling cascade through phospholipases (PLC and PLD). *PAL genes* expression is, in part, under the control of SA and is mediated by second messengers such as IPn (dashed black arrow). These second messengers may increase Ca^2+^-dependent signaling (solid black arrow), which results in the regulation of transcription factors or protein kinases (dashed black arrow), and therefore causing an increase in the expression levels of phenylalanine ammonia–lyase (PAL) genes. However, in the presence of U73122 or neomycin (inhibitors of PLC signaling), the levels of DAG and IPn (second messengers) are reduced (solid red arrow), which leads to the alteration of intracellular Ca^2+^ levels (dashed red arrow) that may affect the accumulation of PAL mRNA. The responses of cells to SA, result in the production of second messengers, such as DAG, IP_n_ or PA, generated from the phospholipid-signaling pathway and are involve in the regulation of PAL activity and in the production of vanillin. 

 represents decrease of PIP_2_ in the presence of neomycin. 

 represents increase of PIP_2_ due to the inhibition in the enzymatic activity of the PLC by U73122.

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
