# Peer review of "Phospholipid Signaling Is a Component of the Salicylic Acid Response in Plant Cell Suspension Cultures"

_ijms, 2020, doi:10.3390/ijms21155285_

Round 1
Reviewer 1 Report
1) Generally, the connection between SA and PL signalling is not very evident and/or based on indirect proofs. Therefore it was suggested not to use speculative consequences.
2) Reconsider the structure of the paper in connection with abiotic versus biotric stress.
Page2 Title 2 indicates abiotic signalling (2. Salicylic acid: an essential regulator to mitigate abiotic stress in plants) but 2.1. is mainly about biotic stress (2.1. Salicylic acid signaling through SA-binding proteins.). Its fine to compare these biotic and abiotic stresses but the subtitle should not counteract with title 2. Please, rename or at least modify titles/subtitles.
In connection with methyl salicylate esterase (SABP2) it was reported that MeSA does not play a role in Arabidopsis resistance response, especially not in systemic one. See Attaran et al., 2009 The Plant Cell, Vol. 21: 954–971.
Page 3 Under 2.2. Importance of SA in crops: abiotic signals/stresses are considered repeatedly (as a continuation of title 2?).
Page 5 Do not use "we".
In 2011, our research group performed the in vitro enzyme assay of microsomes extracted from
223 C. chinense Jacq. suspension cells 30 min after SA treatment. We also found that SA provoked an
224 increase in lipid kinase activities leading to PI4P and PIP2 and a decrease in PI-PLC activity [65]. This
225 finding strongly suggests that lipid kinases and PIP2 are components of a SA signaling pathway and
226 that the activation of this pathway is necessary for the SA response.
Page7 This paragraph is very speculative.
Although the role of those proteins has been well described in
310 studies of the response to ABA, could the mechanism be the same in SA signaling? Unfortunately,
311 we are aware of a single report of a PA binding protein in response to SA. In that report, the authors
312 demonstrated that PA added to the leaves in A. thaliana could prevent the effect of Saw, leading to
313 the depolymerization of actin filaments by binding to the capping protein [103].
Page 8. Please, add reference and/or data.
To examine the possible role of PLD/PA in the transcription of PAL genes in Capsicum, we treated
358 the suspension cells with both SA and 1-butanol, an agent that reduces the levels of PA. We showed
359 that the addition of 1-butanol plus SA had a significant effect on PAL mRNA. Thus, PAL mRNA is
360 not regulated by SA at the transcriptional level, at least in C. chinense
Subtile 6. Connections between SA and phospholipid signaling under aluminum stress. There is rather putative connection but not biochemical/genetic experimental proof.
Author Response
Response to Reviewer 1 Comments
Point 1: Generally, the connection between SA and PL signaling is not very evident and/or based on indirect proof. Therefore, it was suggested not to use speculative consequences.
Response 1: The authors appreciate the reviewer's suggestion. The document has been thoroughly revised to eliminate speculative sentences between SA and phospholipid signaling.
Point 2: Reconsider the structure of the paper in connection with abiotic versus biotic stress.
Page 2 Title 2 indicates abiotic signaling (2. Salicylic acid: an essential regulator to mitigate abiotic stress in plants) but 2.1. is mainly about biotic stress (2.1. Salicylic acid signaling through SA-binding proteins.). Its fine to compare these biotic and abiotic stresses but the subtitle should not counteract with title 2. Please, rename or at least modify titles/subtitles.
Response 2: We appreciate the observations made by the reviewer. The title of page 2, has been modified. Now it is “Salicylic acid: an essential regulator to mitigate biotic and abiotic stress in plants” (Line: 61).
Point 3:
In connection with methyl salicylate esterase (SABP2) it was reported that MeSA does not play a role in Arabidopsis resistance response, especially not in systemic responses. See Attaran et al., 2009 The Plant Cell, Vol. 21: 954–971.
Response 3: The authors agree with the reviewer´s observation. Endogenous MeSA levels increase with pathogenic attack in plants. In systemic tissues, MeSA is converted back to SA to induce resistance (Tripathi et al 2019) while in nonsystemic tissues (in Arabidopsis), methyl esterase activity is not dispensable, as indicated by Attaran et al. (2009). This valuable observation has already been included in the revised document (Lines:91-92).
Point 4: Page 3 Under 2.2. Importance of SA in crops: abiotic signals/stresses are considered repeatedly (as a continuation of title 2?).
Response 4: Thank you for the observation. The subtitle of section 2.2 was revised to “SA an essential regulator to mitigate biotic and abiotic stress in plants”, and this section was rewritten to briefly discuss the advances in research on SA signaling in plants under abiotic stress (Lines:135-155).
Point 5: Page 5 Do not use "we":
In 2011, our research group performed the in vitro enzyme assay of microsomes extracted from C. chinense Jacq. suspension cells 30 min after SA treatment. We also found that SA provoked an increase in lipid kinase activities leading to PI4P and PIP2 and a decrease in PI-PLC activity [65]. This finding strongly suggests that lipid kinases and PIP2 are components of a SA signaling pathway and that the activation of this pathway is necessary for the SA response.
Response 5: We appreciate the observation. The word “we” has been removed from the paragraph as suggested by the reviewer and was replaced with “The results showed” (Line: 264).
Point 6: Page 7 This paragraph is very speculative.
Although the role of those proteins has been well described in studies of the response to ABA, could the mechanism be the same in SA signaling? Unfortunately, we are aware of a single report of a PA binding protein in response to SA. In that report, the authors demonstrated that PA added to the leaves in A. thaliana could prevent the effect of Saw, leading to the depolymerization of actin filaments by binding to the capping protein [103].
Response 6: We appreciate the observation. The paragraph was rewritten so that it is not read as a speculation of the authors. The paragraph was replaced with “The role of those proteins has been well described in studies on the response to ABA; however, studies focused on the identification of some PA binding proteins in response to SA are scarce. There is a study published by Matousková et al. 2014 demonstrating that the negative effect of SA on actin dynamics in Arabidopsis seedlings was abolished by binding of the capping protein (CAP) with PA [88]. Research into the exact functions of PA in SA-triggered signaling is one of the most interesting frontiers of research in plant cells [103]” (Lines: 344-349).
Point 7: Page 8. Please, add reference and/or data.
“To examine the possible role of PLD/PA in the transcription of PAL genes in Capsicum, we treated the suspension cells with both SA and 1-butanol, an agent that reduces the levels of PA. We showed that the addition of 1-butanol plus SA had a significant effect on PAL mRNA. Thus, PAL mRNA is not regulated by SA at the transcriptional level, at least in C. chinense”.
Response 7: We apologize for not including data in that paragraph. In relation to this paragraph, these are results obtained that will be included in a research article that is still being written. This why "[110] (Rodas-Junco et al. In preparation)" was added in the paragraph (Lines: 400-402).
Point 8: Subtitle 6. Connections between SA and phospholipid signaling under aluminum stress. There is rather putative connection but not biochemical/genetic experimental proof.
Response 8: Thank you for the observation. This section has been eliminated from the review to avoid confusion.
Reviewer 2 Report
Some chapters of this review are too general but the others are too specific. For example Chapter “2.2. Importance of SA in crops”: half of the chapter is about general effects of SA (“crucial roles in the regulation of plant development processes and signaling networks and is directly or indirectly involved in a wide range of responses that contribute to stress tolerance in plants“) and in the second half only about tissue cultures.
It can be seen from the all review that authors work on cell cultures because some of the chapters only about it and whole plants or plant organs are not mentioned. I am exactly sure that the effects of SA are not the same in the cell cultures in whole plants even it can be different in the same plants in the different organs. The cell culture parts are fine but the others are too general containing not too much recent relevant information. I think it should be either focused to the cell cultures or discuss every aspect from cell cultures/plant organs/whole plants. In the first case the title should be modified “Phospholipid Signaling is a component of the Salicylic Acid response in Plant Cell Suspension Cultures” instead of “Phospholipid Signaling is a component of the Salicylic Acid response in Plants”.
Chapter 6.: I think it is not necessary for this review. Why only the Al stress? The other abiotic stresses can be also important. On the other hand, half of this chapter is mainly about “heavy” metals but the metals in row 363 are not “heavy” metals even not all of them are toxic ones. Later toxic metals are mentioned such as As, Cd, Hg etc.
Rows: 400-407: Effects of Al stress on phospholipid signalling
Is it specific to Al or can other metals also induce it?
Notes:
- Abbreviations should be defined at the first appearance then used it consistently.
- SA concentrations: row 176: 25 uM; row 179: 250 uM. Is it the same experiment or not? It is not clear. Row 284: the concentration of SA is not written. I think it is important because the protective effect can be seen at different concentration in different plant species and it is not a wide range. Too much SA can be harmful to the plants especially at higher temperatures.
Author Response
Point 1: Some chapters of this review are too general, but the others are too specific. For example, Chapter “2.2. Importance of SA in crops”: half of the chapter is about general effects of SA (“crucial roles in the regulation of plant development processes and signaling networks and is directly or indirectly involved in a wide range of responses that contribute to stress tolerance in plants“) and in the second half only about tissue cultures.
It can be seen from the all review that authors work on cell cultures because some of the chapters only about it and whole plants or plant organs are not mentioned. I am exactly sure that the effects of SA are not the same in the cell cultures in whole plants even it can be different in the same plants in the different organs. The cell culture parts are fine but the others are too general containing not too much recent relevant information. I think it should be either focused to the cell cultures or discuss every aspect from cell cultures/plant organs/whole plants. In the first case the title should be modified “Phospholipid Signaling is a component of the Salicylic Acid response in Plant Cell Suspension Cultures” instead of “Phospholipid Signaling is a component of the Salicylic Acid response in Plants”.
Response 1: The authors appreciate the suggestion. The revised manuscript focuses primarily on discussing aspects of phospholipid signaling in response to SA in plant cells. This is why the title has been changed to "Phospholipid Signaling is a Component of the Salicylic Acid Response in Plant Cell Suspension Cultures”, as suggested.
Point 2: Chapter 6.: I think it is not necessary for this review. Why only the Al stress? The other abiotic stresses can be also important. On the other hand, half of this chapter is mainly about “heavy” metals but the metals in row 363 are not “heavy” metals even not all of them are toxic ones. Later toxic metals are mentioned such as As, Cd, Hg etc.
Rows: 400-407: Effects of Al stress on phospholipid signalling
Response 2: The authors appreciate the reviewer's suggestion. This section has been eliminated from the review to avoid confusion.
Point 3: Is it specific to Al or can other metals also induce it?
Response 3: As stated in point 2, this section regarding Al stress has been eliminated from the review.
Point 4: Abbreviations should be defined at the first appearance then used it consistently.
Response 4: We appreciate your comment. The format of the entire manuscript has been carefully corrected with respect to the appropriate use of abbreviations.
Point 5: SA concentrations: row 176: 25 uM; row 179: 250 uM. Is it the same experiment or not? It is not clear.
Response 5: We apologize for the mistake in the SA concentration in lines 176 and 179. The correct concentration should be 25µM not 250µM. This has already been corrected in the manuscript (Line: 217).
Point 6: Row 284: the concentration of SA is not written. I think it is important because the protective effect can be seen at different concentration in different plant species and it is not a wide range. Too much SA can be harmful to the plants especially at higher temperatures.
Response 6: The reviewer has made an excellent observation, and we agree. The concentration of SA used in that investigation was 200µM, which has already been included in the manuscript (Line: 221).
Reviewer 3 Report
This is a thorough, investigative work on a significant subject. It covers a lot of intriguing thoughts on an exceptionally intricate framework.
Unfortunately, the paper is fairly inadequately composed.
I suggest the following tasks to make the paper publishable:
(a) A careful linguistic audit ought to be done to dispense with the syntax problems and the uncertain and entangled wording. (b) The roles of salicylic acid in plant physiology have been extensively studied during the last decades and discussed in numerous reviews and books: I think Section 2 of the paper could be reduced to a couple of paragraphs. (c) Since the authors are very active in the field of the study and published a number of papers, they should clearly express in the Abstract and the Conclusions what is novel and unique in this paper of theirs (approach and findings). They also need to identify the advances they made since the publication of their book section titled Salicylic Acid and Phospholipid Signaling. (d) The Discussion segment lacks originality: findings from a large number of works are merely listed (correctly, though) yet they are not assessed critically in the light of the latest discoveries (made by the authors or researchers from other laboratories). (e) The Conclusions part is ambiguous and should be improved. (f) The number of references needs to be reduced: instead of citing many original papers, recent reviews could be referred to (including the author’s review mentioned in the paragraph above).
By and large, I accept that the paper, regardless of its shortcomings in wording and presentation, is certainly publishable.
Itemized opinion
Quality of the writing style: poor / medium
Popularity of the subject: good
Topic is significant: good
Appropriateness for the Journal: good
Adequacy of literature review: good
Quality of research design: good
Problem(s) adequately addressed: good
Adequacy of data analysis: good
Manuscript integrative in regard to theory and practice: not applicable
The major relevant references included: good
Contains new findings / ideas: medium
Uniqueness of the contribution: medium
Conclusions are carefully drawn, supported, explained, and tied to existing literature: medium
Practical / managerial significance: not applicable
Manuscript is well-organized and clearly written: poor / medium
The findings are adequately related to current literature: medium
Overall evaluation: medium / good
Recommendation for review: revisions required
Author Response
Response to Reviewer 3 Comments
This is a thorough, investigative work on a significant subject. It covers a lot of intriguing thoughts on an exceptionally intricate framework. Unfortunately, the paper is fairly inadequately composed. I suggest the following tasks to make the paper publishable:
Point 1: A careful linguistic audit ought to be done to dispense with the syntax problems and the uncertain and entangled wording.
Response 1: The authors thank the reviewer for the comment. Sections have been rewritten to improve the writing and to avoid any syntax problems. The manuscript has been edited for English grammatical errors as suggested by the reviewer.
Point 2: The roles of salicylic acid in plant physiology have been extensively studied during the last decades and discussed in numerous reviews and books: I think Section 2 of the paper could be reduced to a couple of paragraphs.
Response 2: The reviewer has made an excellent observation. Although several authors have published excellent reviews about the role of SA in plants, we consider it important to briefly address SAPBs in the molecular scenario of SA perception and signaling in response to biotic stress. Likewise, section 2.2, which was renamed "Salicylic acid: an essential regulator to mitigate biotic and abiotic stress in plants", it was restructured to address the importance of SA in improving tolerance to abiotic stress by controlling the metabolic processes of phytohormones in plant cells (Lines: 135-155).
Point 3: Since the authors are very active in the field of the study and published a number of papers, they should clearly express in the Abstract and the Conclusions what is novel and unique in this paper of theirs (approach and findings). They also need to identify the advances they made since the publication of their book section titled Salicylic Acid and Phospholipid Signaling.
Response 3: The authors thank the reviewer for this comment. The abstract and conclusions has been rewritten to clarify the scope of the research linking the phospholipid and SA-mediated signaling pathways in plant cells.
Point 4: The Discussion segment lacks originality: findings from a large number of works are merely listed (correctly, though) yet they are not assessed critically in the light of the latest discoveries (made by the authors or researchers from other laboratories).
Response 4: The authors appreciate the reviewer suggestion. The authors have improved the critical analysis in the discussion section as suggested.
Point 5: The Conclusions part is ambiguous and should be improved.
Response 5: The authors appreciate the suggestion. The authors have improved the conclusions of the manuscript by briefly highlighting important aspects that should be explored in the context of SA-phospholipid signalling (Lines: 406-430).
Point 6: The number of references needs to be reduced: instead of citing many original papers, recent reviews could be referred to (including the author’s review mentioned in the paragraph above). By and large, I accept that the paper, regardless of its shortcomings in wording and presentation, is certainly publishable.
Response 6: The authors appreciate the suggestion. The amount of references has been reduced as suggested.
Round 2
Reviewer 1 Report
I appreciate most of the modifications indicated in the reponse letter.
There are more, than many SABPs in different plants. Response 3 needs further corrections in connection with SAPB2 as the interpretation of the paper of Attaran et al. (2009) is misleading and incorrect at present. The point is the methyltransferase gene and the role of MeSA in SAR as indicated below:
"T-DNA insertion lines defective in expression of a pathogen-responsive SA methyltransferase gene are completely devoid of induced MeSA production but increase systemic SA levels and develop SAR upon local P. syringae inoculation. Therefore, MeSA is dispensable for SAR in Arabidopsis, and SA accumulation in distant leaves appears to occur by de novo synthesis via isochorismate synthase."
Point 7. I could not understand your conclusion. The double treatment was effective but SA or 1-butanol not on PAL mRNA level? 1-butanol and/or SA influenced PA level too? I would omit non-published results.
Author Response
Response to Reviewer 1 Comments, Round 2.
Point 1: There are more, than many SABPs in different plants. Response 3 needs further corrections in connection with SAPB2 as the interpretation of the paper of Attaran et al. (2009) is misleading and incorrect at present. The point is the methyltransferase gene and the role of MeSA in SAR as indicated below:
"T-DNA insertion lines defective in expression of a pathogen-responsive SA methyltransferase gene are completely devoid of induced MeSA production but increase systemic SA levels and develop SAR upon local P. syringae inoculation. Therefore, MeSA is dispensable for SAR in Arabidopsis, and SA accumulation in distant leaves appears to occur by de novo synthesis via isochorismate synthase."
Response 1: We appreciate the valuable observation made by the reviewer. This observation has been included in the revised manuscript (Lines: 88-92).
Point 2:
Point 7. I could not understand your conclusion. The double treatment was effective but SA or 1-butanol not on PAL mRNA level? 1-butanol and/or SA influenced PA level too? I would omit non-published results.
Response 2: We appreciate the observation and we also apologize for not making that paragraph clear. To examine the possible role of the PLD/PA pathway, we treated cell suspensions with SA, 1-butanol and 1-butanol + SA. Our results showed that the treatment of 1-butanol + SA increased the levels of CchPAL1 and CchPAL5 mRNA in C. chinense cell suspensions, therefore we infer that in the transcriptional regulation of these putative genes in response to SA, the PLD/PA is not involved. Regarding PA levels, in 2015 our group reported that PA levels increased in response to SA while the treatment of cells with 1-butanol + SA generated a decrease in PA levels (Rodas-Junco et al. 2015). Finally, it is important to mention that the paragraph has been rewritten and "non-published results" have been removed from the manuscript as suggested (Lines: 378-380).
Reviewer 2 Report
Requested changes have been made.
Author Response
We appreciate and thank the referee´s feedback. His suggestions helped to improve the document.